# Effect of Powder Recycling on the Organization and Mechanical Properties of GH4169 Alloy by Laser Metal Deposition

**Haibo Zhang, Jieshuai Li and Yingqiu Li \***

School of Mechanical Engineering, Northeast Electric Power University, Jilin 132012, China;
2202000727@neepu.edu.cn (J.L.)
\* Correspondence: 20213046@neepu.edu.cn

**Abstract:** The purpose of this research is to prepare GH4 169 alloy specimens by laser metal deposition, by investigating the changes in powder morphology, powder particle size, and elemental content during the cycling process. As well as the pore defects and microstructure of deposited samples prepared from recycled powder, we analyzed the changes in powder properties during the cycling process and the effects of using recycled powder on the organization and properties of LMD-deposited specimens. It was shown that the average particle size of the powder increased with the increase in the size of powder recycling, from 59.861 μm in the original powder to 64.144 μm after four cycles, with the phenomenon of powder burnout and satellite ball. The elemental content of the powder changed with the increase in the number of cycles, among which the elemental content of Nb increased significantly from 4.31 wt% of the original powder to 7.97 wt% after four cycles, the proportion of Laves phase in the deposited samples increased, the porosity and pore size increased, the tensile strength of the specimen decreased from 1046 MPa of the original powder to 936 MPa, the tensile strength decreased by 10.5%, and the elongation was increased to 27% from 11% of the original powder. Powder recycling can lead to powder aging and reduce the mechanical properties of the laser metal deposited formed parts.

**Keywords:** laser metal deposition; powder recycling recovery; GH4169 alloy





## 1. Introduction

High-temperature alloy is an alloy that works under conditions above 600 °C, can withstand large stresses, and has excellent high-temperature properties such as corrosion resistance and oxidation resistance. It has an irreplaceable role in aero engines and high temperature gas turbine hot end components. GH4169 alloy is a precipitation-reinforced nickel-based high-temperature alloy with good corrosion resistance, creep resistance, tensile strength, and other properties in the temperature range of −253 to 650 °C. It is widely used in aerospace engine blades, turbine disks, and other parts, and is the high-temperature alloy with the highest application ratio [1–4].

Laser metal deposition (LMD) is an additive manufacturing technology that introduces powder into a molten pool generated by a high-power laser to form complex parts layer by layer, which has the characteristics of near-net forming and high density of formed parts and is widely used in aerospace and medical industries [5,6]. Compared with traditional processing and manufacturing methods, LMD-prepared parts have higher densities under suitable process parameters, due to the high cooling rate of laser melting deposition, smaller grains can be obtained, which results in higher strength and lower ductility of the deposited samples. Compared with traditional processing methods, the mechanical properties of laser additive manufacturing can be increased by 10%–50% [7]. However, the current utilization rate of laser metal deposited powder is about 75%, and there is inevitably a waste of unused powder during the deposition process. The powder is rarely reused in the production process due to the lack of specifications for the treatment of

recycled powder. In order to reduce the cost of additive production and improve the utilization of powder for additive manufacturing, as well as to reduce the impact of powder recycling on the performance of parts, there are more experimental studies on the effect of powder recycling on the performance of formed parts during the melting or additive manufacturing process. L.C. Ardila [8] et al. found that the mechanical properties of IN718 powder specimens were not significantly different from those of the original IN718 powder after its recovery by multiple SLM utilization. Mario Renderos [9] et al. studied the presence of magnetic impurities in IN718 powder after sieving of the residual powder after multiple laser cladding. Liu Baoyuan [10] studied the recovery of TC4 powder several times under the SLM process, and studied the effect of the recovery process on TC4 powder parameters and mechanical properties of formed specimens, and found that there was no significant change in powder sphericity after 14 cycles, powder fluidity increased, and mechanical properties such as tensile and fatigue were basically the same under different cycle times. Seyda V [11] et al. also studied the recycling of TC4 alloy powder in SLM process, which also found that the average powder particle size increased and the powder flowability was improved with the increase of the number of powder recycling. Y. Y. Sun [12] compared in detail the differences between new and recycled powders in the new titanium alloy in terms of powder characterization parameters, the study included powder granularity, powder size, loose packing density, rest angle, etc. The results showed that several properties of the powder changed significantly during the recycling process. The study for the preparation of GH4169 alloy by LMD is still in the preliminary stage. The overall study on the effect of powder recycling process on powder parameters and mechanical properties of formed parts of GH4169 alloy under LMD process is still less, and there are still few reports on the effect of powder recycling process on the precipitation phase of LMD deposition samples.

This research investigates the effects of different powder recycling times on the morphology, powder composition, microstructure and mechanical properties of GH4169 alloy powder prepared by laser metal deposition (LMD) to verify the feasibility of reusing GH4169 alloy powder prepared by LMD and investigating the mechanism of the effect of using recycled powder on the properties of GH4169 alloy prepared by LMD. This helps to establish the treatment specification for the recycled powder in the LMD process, reduce the cost of LMD production, improve the utilization rate of the powder, study and elaborate on the effect of the recycled powder on the performance of the parts, in order to provide the corresponding theoretical guidance for the recycling of the powder.

## 2. Materials and Methods

The aerosol preparation of GH4169 alloy powder used in this experiment was provided by Höganäs, powder composition as shown in the Table 1, in order to reduce the impact of the coefficient of thermal expansion of the sample and other physical properties on the deposition properties, this experiment uses forged GH4169 alloy as the sample, the powder is dried in a vacuum drying oven at 140 °C for 1 h before the experiment to reduce the effect of water vapor in the powder on the properties of GH4169 alloy, the experimental process with sandpaper was used to remove the oxide layer of the sample and ultrasonic cleaning was performed with acetone.

**Table 1.** GH4169 alloy powder composition (Wt/%).

| Si | Cr | Mo | Co | Nb | Al | Ni | Fe |
|------|------|------|------|------|------|------|------|
| 0.23 | 18.3 | 2.8 | 0.79 | 5.4 | 0.45 | 54.2 | Bal. |

The laser metal deposition process is shown in Figure 1. The laser metal deposition system used in this experiment consists of IPG Photonics fiber laser (YLS-4000), GTV powder feeder, circulating water cooling system prepared by Sanhe Tongfei Refrigeration Co., Ltd. (Sanhe, China) coaxial powder feed nozzle and five-axis quadruple-action argon-filled chamber CNC machine prepared by Etxlux. The argon-filled chamber can ensure and main-

tain the volume fraction of water oxygen content in the chamber less than $1 \times 10^{-5}$ ppm, reducing the effect of oxygen content in the environment on the performance of GH4169 alloy. The experimental parameters used were defocusing +17 mm, powder feeding rate 25 g/min, laser power 1000 W, deposition scan rate 1000 mm/min, and deposition of 100 mm × 15 mm × 10 mm samples. During the experiment, a metal tray was placed on the operating table to collect powder, and the collected powder was sieved with a 200-mesh sifter to remove the excess residue and oxide skin, and then put into a vacuum drying oven at 140 °C for 1 h. The above steps were repeated four times to obtain the deposited samples of GH4169 alloy prepared by LMD after cycling 0–4 times.

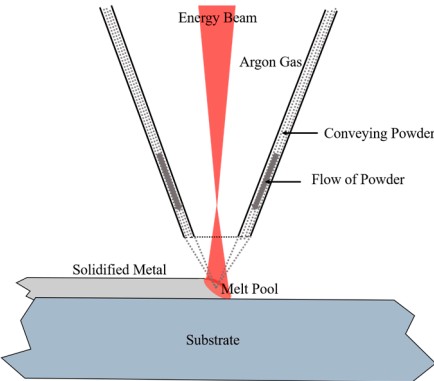

**Figure 1.** Schematic diagram of laser metal deposition.

Powder morphology, powder composition and deposited layer microstructure were analyzed using scanning electron microscopy (SEM, FEI Company, Hillsboro, OR, USA). The original powder (H0) and the four recovered powders (H1–H4) were characterized using a Winner 2000ZDE laser particle analyzer, laser particle size measurement range of 0.1–300 μm, measurement resolution 0.001 μm. The microstructure of the five deposited samples was analyzed by Zeiss Axio Observer optical microscope, the etchant used was copper sulfate etchant, and the corrosion test was conducted for 1 min. After LMD processing, the sample was prepared with a specific geometry as shown in Figure 2, and tensile testing is performed using a zwick-z100 tester. The tensile properties of recycled powder were tested and analyzed after different cycles.

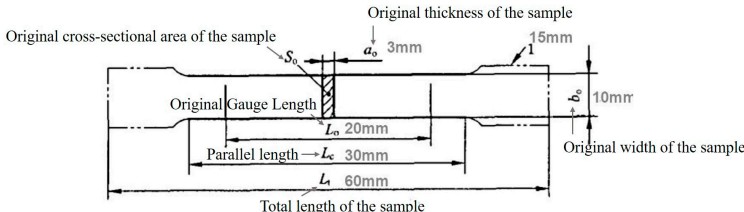

**Figure 2.** Tensile specimen geometry.

## 3. Analysis and Discussion

### 3.1. Powder Characterization for Different Recycling Times

Whether or not the recovered powder can be reused depends on the sphericity, powder size and powder composition. Figure 3 shows the morphology of the original powder and the GH4169 alloy powder after the 1st, 2nd, 3rd, and 4th reuse. It can be seen from Figure 3 that the new powder has a regular spherical shape, and the surface of the powder is smooth without adhesion, while the recycled powder has apparent traces of powder burn, and some particles are no longer spherical. The powder has adhesion, and a spherical satellite shape, but the overall sphericity of the powder after the four recycling is good. Figure 4 shows the powder's particle size distribution under different cycle times. The largest proportion of the original powder particles is 60–65 μm, with the increase in cycle

times, the largest proportion of the fourth recovery powder particles is 78–85 µm, and its average particle size increases from 59.861 to 64.144 µm after four cycles, at the same time the particle size distribution map has more obvious enrichment peaks. Figure 5 shows the D90, D50, and D10 values in different powder samples, where D90 indicates that 90% of the particles in the new powder have a particle size less than 78.309 µm, and D50 indicates that 50% of the particles in the new powder have a particle size less than 60.288 µm, while the D90 of the powder is 84.340 µm and D50 is 64.530 µm after four cycles. The results show that the powder particle size increases after recycling, mainly due to the smaller powder melting under the action of laser energy and adhering to the larger powder particles, making the powder volume fraction increase. In the process of laser metal deposition, the powder is transported to the focus of the laser beam under the act of the carrier airflow. Therefore, the smaller particles of the powder are more likely to converge to the center of the powder stream under the act of the airflow, thus are more easily irradiated by the laser energy to form a melt pool. In comparison, the larger powder particles are easily scattered at the periphery of the powder stream, which will be recovered.

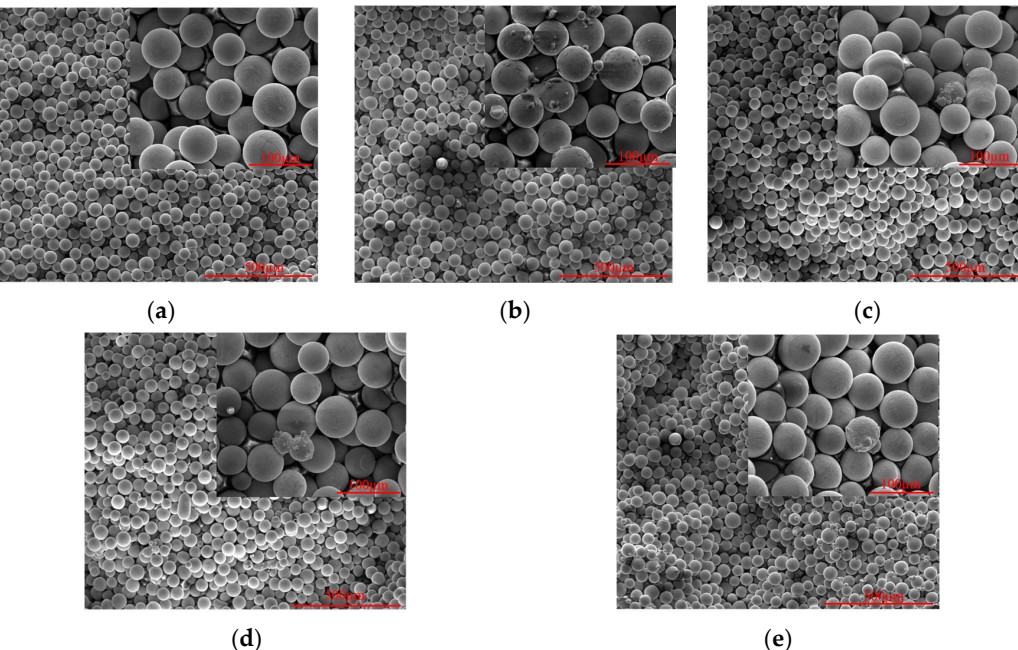

**Figure 3.** SEM morphology of the powder. (**a**) new powder; (**b**) one time recovery powder; (**c**) two times recovery powder; (**d**) three times recovery powder; (**e**) four times recovery powder.

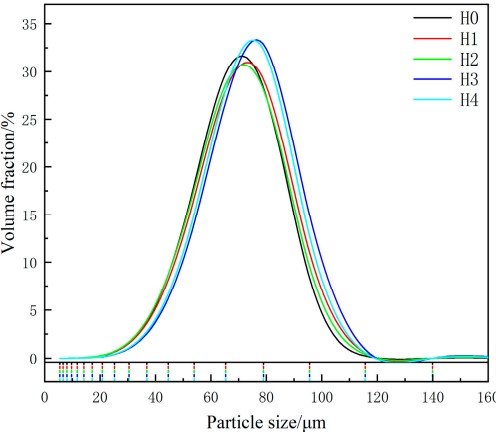

**Figure 4.** Particle size distribution of recycled powder.

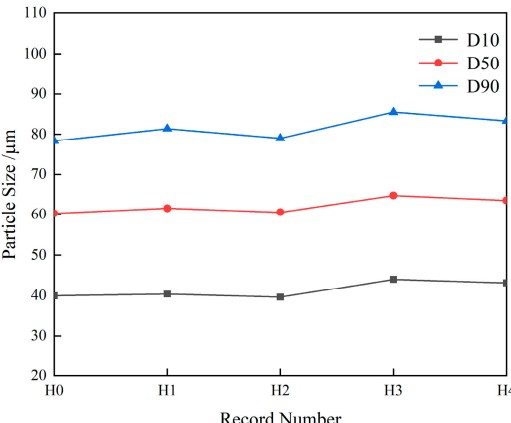

**Figure 5.** D values of particles.

The effect of the recycling process on the powder composition was investigated by using Energy Dispersive Spectrometer (EDS, Ametek Trading Co., LTD, Shanghai, China) on the recycled powder. As shown in Figure 6, Nb elemental content shows an increasing trend with the increase in powder recycling, from 4.31 wt% of the original powder to 7.97 wt% after four cycles, probably due to the strong antioxidant capacity of Nb, which results in a smaller loss in the recycling process, leading to an overall increase in its mass percentage. Therefore, changes in element content in GH4169 alloy can significantly affect the organization and properties of the LMD-deposited sample. The GH4169-deposited samples precipitated phase in laser metal deposition is mainly the Laves phase, a hazardous phase [13]. An increase in the content of Nb elements will lead to a rise in the degree of Nb segregation, which will affect the Laves phase content and, therefore, may affect the properties of deposited samples.

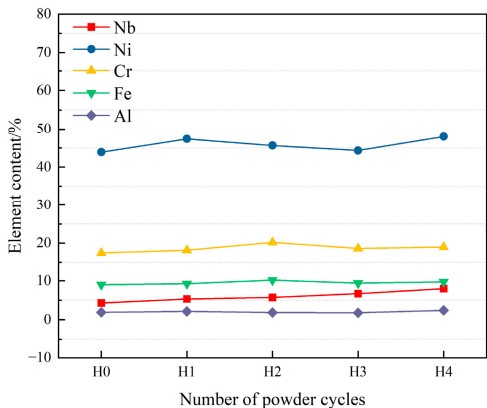

**Figure 6.** Elemental content of recycled powder.

### 3.2. Microstructure Analysis of Powder Deposition Samples with Different Recycling Times

The morphological and compositional analysis of the powder indicates that the powder recycled will not significantly change the characteristics of the powder, and the laser metal deposition is a process of forming parts by layer accumulation of molten powder [14], so it is necessary to study the effect of powder recycling on its deposition. The deposited sample nuggets obtained under different cycle times of powder were cut and ground and polished to obtain metallographic micrographs after different cycle times of recovery, and the results are shown in Figure 7. The number of recycling increases, the deposited layers still show good metallurgical bonding, and the microstructure of deposited samples is mainly composed of dendrites. The experiment uses orthogonal scanning for layer-by-layer stacking, so the dendrite growth direction is not continuous, which is because the laser scanning direction of the former layer of the orthogonal scanning method is perpendicular

to the laser scanning direction of the latter layer, which makes the thermal dissipation of the deposited layer change, and the continuous growth of columnar crystals is inhibited.

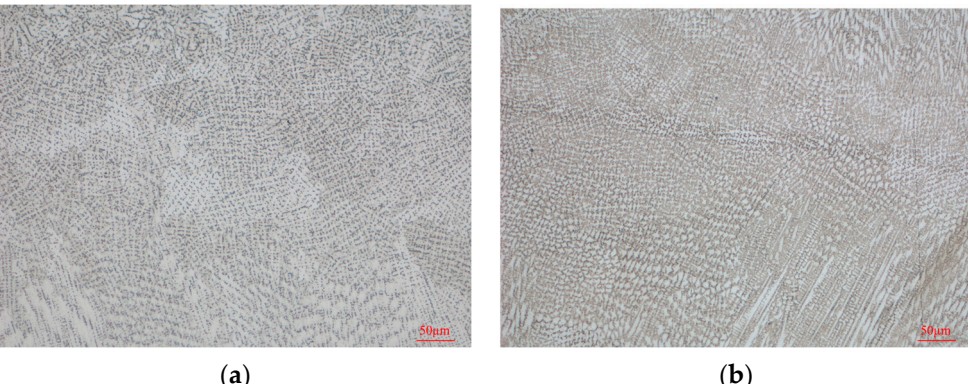

| (**a**) | (**b**) |

**Figure 7.** Metallographic photograph of powder deposition samples. (**a**) New powder (H0) deposited metallographic photos; (**b**) four times recovery (H4) deposited metallographic photos.

Compared with the new powder deposited samples, the H4 specimen has more columnar crystals, showing overgrown samples. The average particle size of the recycled powder increases, the gap between the powders increases, and the laser energy is more easily reflected between the powder particles during the irradiation process, which increases the absorption of laser energy by the powder, and the subcooling of the melt pool decreases, and the dendrites grow more fully [15].

The SEM observation indicates that there is an obvious white chain-like precipitation phase between the dendrites, as shown in Figure 8a,b, which may be the Laves phase in combination with the existing studies [16], and the EDS analysis of the matrix (Spectrum1) and the precipitation phase (Spectrum2) in the four recovered deposition samples is shown in Figure 8c,d. The Nb element content is as high as 24.8 wt%, which is significantly higher than the matrix Nb content, and the Nb deviations are serious, which can be identified as the Laves phase. The Laves phase in the deposited samples of the unrecycled powder was in the form of chains of different lengths, and the Laves phase in the deposited samples of the powder was in the form of a continuous grid after four recycling processes. The primary dendrite arm spacing (PDAS) and Laves phase volume fraction of each specimen were analyzed by the Renyi entropy color threshold method of ImageJ image processing software, and both measurements were averaged 20 times, the results showed that the Laves phase volume fraction increased from 6.12% of the original powder to 8.35% after four cycles. Indicating that the proportion of the Laves phase in the deposited samples obtained after powder recycling increased, It has been shown that the Laves phase is brittle, which is usually a vulnerable area for crack budding and expansion, and will reduce the mechanical properties of the deposited samples [17–19] The primary dendrite arm spacing of the specimen increased from 6.32 μm in the original powder to 8.47 μm after four cycles. The subcooling degree is the main factor affecting the dendrite spacing. As the subcooling degree decreases, the cooling rate decreases and the nucleation rate decreases, while the growth time of the dendrite increases and the dendrite arm spacing increases accordingly. According to related studies [20,21] the relationship between the dendrite arm spacing ($\lambda$) and the cooling rate ($\dot{T}$) is as follows:

$$\lambda = 104.47 \times \dot{T}$$

The surface of the recycled powder particles is burned and accompanied by satellite spherical morphology, which increases the granularity of the powder and the surface area of the powder, making the powder absorb more laser energy, the subcooling of the melt pool decreases, as mentioned above, the cooling rate decreases, the Nb element diffuses

more easily between the dendrites, and the Nb segregation increases, leading to an increase in the dendrite spacing and the volume fraction of the Laves phase.

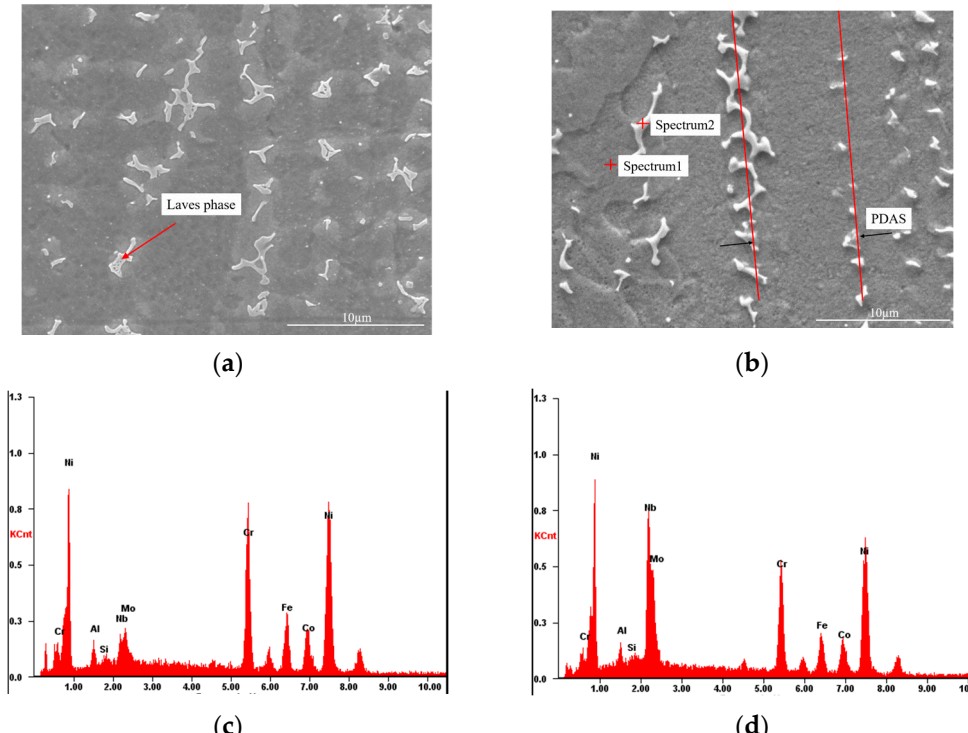

**Figure 8.** SEM photo of powder deposition samples. (**a**) SEM photo of the deposited sample of new powder (H0); (**b**) SEM photo of the deposited sample of four times recovered (H4); (**c**) Sample EDS; (**d**) Laves phase EDS.

There are two main reasons for the formation of pores in the laser metal deposition process, one is due to the deposition process of the protective gas due to the rapid condensation characteristics did not have time to escape from the melt pool and the formation of pore defects, or due to the laser energy makes the alloy powder vaporization, metal vapor condensation process did not escape in time to form pore defects, as shown in Figure 9 (pore 1); Second, inclusions or insufficient pressure in the two-phase porridge area of the melt pool cause irregularly shaped keyhole defects, as shown in Figure 9 (pore 2) [22–24]. After recycling, the powder particles are easily adhered to each other to form a satellite sphere shape, when laser energy is applied to the powder particles, the gas between the powder interstices is brought into the melt pool, which is confined in the melt pool due to the fast cooling property and forms irregular keyhole pores together with the surrounding tiny particles that are not completely melted.

In order to observe more easily the porosity variation of the deposited samples obtained from different number of cycles. The microscopic pore size statistics of the longitudinal cross-section of the deposited sample with different powder recycling times were carried out by ImageJ software, and the pore distribution under different recycling times was obtained as shown in Figure 10. As shown in Figure 10, the pore size in the deposited samples of unrecycled powder is less than 80 μm, and the pore size tends to increase with the increase of the number of powder recycling, and the overall number of pores also increases with the increase of the number of powder recycling. All the process parameters used in this experiment are the same, so the change of pore size in the deposited samples is mainly influenced by the number of powder cycles. As the number of cycles increases, the fine powder particles decrease, the average powder size increases, and the number of instances of powder burnout and adhesion increases, thus increasing the chance of pore generation. The increase in the number of powder cycles and powder adhesion lead to an

increase in the number of non-spherical powder particles, which reduces the loose packing density of the powder, and the study [25] shows that when the loose packing density of the powder is low, the gas between the powders may dissolve in the melt pool, and due to the fast cooling characteristics of laser metal deposition, the gas does not escape in time and is confined in the melt pool to form pores. Recovered powder increases the body surface area due to satellite sphere morphology, and results in increased absorption of laser energy by the powder particles. Studies have shown [26] that under the same process parameters, the laser absorption rate affects pore formation, the melt pool temperature increases under high absorption rates, making it easier to form locked hole defects. Changes in pore size and number can significantly affect the mechanical properties of deposited samples.

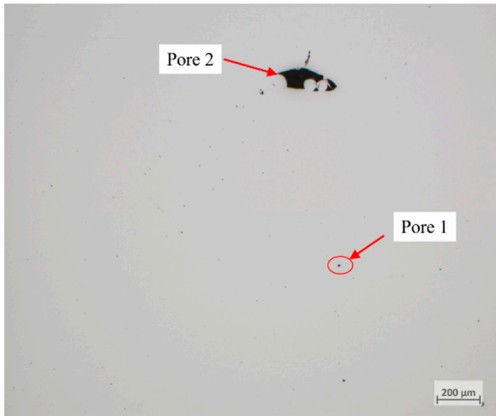

**Figure 9.** Pore defect morphology.

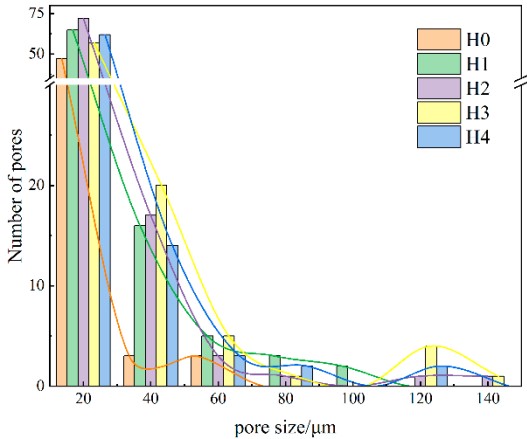

**Figure 10.** Deposited sample porosity.

### 3.3. Mechanical Properties of Powder Deposited Samples with Different Number of Cycles

Microstructure and porosity analysis of the deposited samples with different number of cycles of the powder show that the number of cycles affects the microstructure and internal quality of the deposited samples. However, it is still necessary to test the mechanical properties of the samples to verify that the reuse of the powder may affect the quality of the laser metal deposited formed parts. In this experiment, the effect of the number of cycles on the mechanical properties of the deposited samples was verified by testing the room-temperature tensile properties of powders with different numbers of cycles. Tensile tests were conducted on GH4169 alloy prepared by LMD with different number of powder cycles. Three tensile tests were conducted on GH4169 alloy prepared by each number of powder cycles, and the typical stress-strain curves obtained are shown in Figure 11. It can be seen that the ultimate tensile strength of the deposited sample after recycling is lower than that of the original powder. The average tensile strength of the deposition from

the original powder is 1046 MPa with an elongation of 11%. The average tensile strength of the samples obtained after the powder was recycled four times decreased to 936 MPa, and its elongation increased to 27%, with a 10.5% decrease in tensile strength. It means that the tensile strength of the deposited sample will be reduced by recycling the powder several times. However, the ductility of the deposited samples will be increased, mainly by the more numerous pores in the deposited sample which easily become a source of crack formation and expansion.

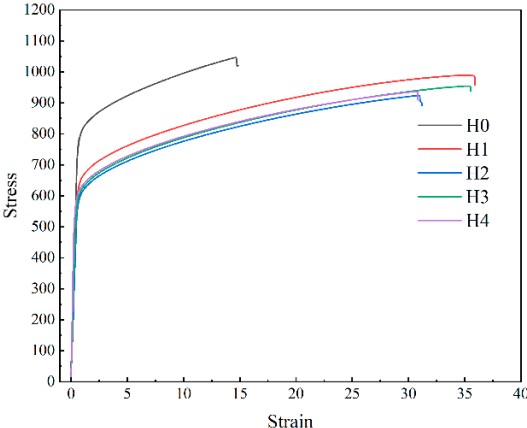

**Figure 11.** Tensile strength of deposited specimens.

## 4. Conclusions

In this research, a comparative study between the new powder and recycled powder of GH4169 alloy was conducted to investigate the effect of the recycling process on the microscopic morphology, microstructure and tensile strength of the powder, and the following conclusions were drawn based on the experimental results:

1.  the recycling process increased the average particle size of the powder, some of the powders were accompanied by burnout bonding, and the content of Nb elements in the powder increased with the increase of the number of cycles.
2.  powder recycling affects the microstructure of the deposited samples, the columnar crystal in the deposited sample is overgrown after recycling, and the percentage of Laves phase in the deposited sample of the recycled powder increases from 6.12% of the original powder to 8.35% after four cycles, the dendrite arm spacing increased from 6.32 μm in the original powder to 8.47 μm after four cycles.
3.  powder recycling affects the number of pore defects in the deposited sample, and the number of pores and pore size in the deposited samples increase significantly after the powder is recycled.
4.  The recycling process reduced the tensile strength of the deposited sample from 1046 MPa of the original powder to 936 MPa after four cycles, but the ductility of the deposited sample increased.

**Author Contributions:** Conceptualization, H.Z.; data curation, J.L.; writing—original draft preparation, J.L.; writing—review and editing, Y.L. All authors have read and agreed to the published version of the manuscript.

**Funding:** This research received no external funding.

**Institutional Review Board Statement:** Not applicable.

**Informed Consent Statement:** Not applicable.

**Data Availability Statement:** Not applicable.

**Conflicts of Interest:** The authors declare no conflict of interest.

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
