# Peer review of "Effect of Powder Recycling on the Organization and Mechanical Properties of GH4169 Alloy by Laser Metal Deposition"

_coatings, doi:10.3390/coatings13030659_

Round 1

Reviewer 1 Report

Dear Authors

I am attaching my comments.

Kind regards

Reviewer

Author Response

Dear Editor

We feel great thanks for your professional review work on our article. As you are concerned, there are several problems that need to be addressed. According to your nice suggestions, we have made extensive corrections to our previous draft, To make it easier for you to quickly find and view the changes, we have underlined the changes in red,the detailed corrections are listed below.

  1. Lines 10 and 11 (abstract) - please explain why the measurement of the diameter of powder spheres with an accuracy of 0.001 micrometer is given.

Response: Powder particle size detection using Laser particle size analyzer, model Winner2000ZDE, its measurement range is 0.1-300 µm, resolution of 0.001µm, recovery powder granularity difference is not large, in order to better reflect the difference in particle size between different cycle times, using a higher precision particle size meter for detection.

  1. After line 20, enter the title of Chapter 1 "Introduction".

Response: As per your suggestion, the quotes have been added to the article and you can view them on line 24.

  1. I propose to change the name of chapter 1 "Experiment" to chapter 2 "Materials and Methods" and renumber the following chapters respectively.

Response: Based on your suggestion, the title of Chapter 2 has been changed to Materials and Methods, which you can see on line 79.

  1. In line 52, please enter the name of the powder manufacturer.

Response: As per your suggestion, the powder manufacturer Höganäs has been added to the article, you can see it on line 81.

  1. The name of the machine and its manufacturer used in the LMD technology, please provide.

Response: The equipment models and manufacturers of the LMD systems included have been added to the article, which you can view in lines 88-92.

  1. Please cite at the end of the title of Table 1, the reference on the basis of which the data on the percentage composition were given.

Response: The elemental content of the powder in your proposed Table 1 was provided by the powder supplier and no other references were cited.

  1. Line 61 - Please enter the full names of IPG and GTV abbreviations.

Response: The full name of IPG has been added to the article, which you can see on line 89. GTV is the German GTV company, not an abbreviation.

  1. Line 75/76 - please write with what accuracy the laser meter used in the experiment measures.

Response: The accuracy of the laser particle size analyzer has been added to the article, which you can see in line 105.

  1. After line 80, please provide the dimensions of the prepared specimens for the static tensile test. It's best to attach a sample drawing. Specify the number of specimens and how they were prepared. Please also provide information on which testing machine the tensile tests were performed.

Response: As suggested by you, the dimensions of the tensile specimen have been added to the article, which you can see in Figure 2 after line 111.

  1. Fig. 2, 5, 6a, and 6b - The text on the microscopic images is hard to read, please enlarge the font or add an appropriate description under the image.

Response: The quality of the images in the article has been optimized, you can see the updated figures 3, 7, 8a, 8b.

  1. Lines 93 and 94 - does the laser gauge measure within 0.001 micrometer? How many diameter measurements were taken to get the average? Please enter the standard deviation or uncertainty of diameter measurements. These values are of great importance in the case of measuring the diameters of powder grains.

Response: Powder particle size detection using Winner2000ZDE model Laser particle size analyzer, its resolution is 0.001um, different cycle times of powder are performed three times particle size detection, in each particle size detection, the instrument calculates the measurement to get the average value, powder particle size measurement deviation depends on the accuracy error of the instrument, Winner2000ZDE laser particle size meter accuracy error < 0.5%.

  1. Fig. 3 - the drawing is a hard read. Please post a better-quality drawing.

Response: The image quality has been optimized, you can view the updated Figure 4.

  1. Line 104 - please enter the full name of the EDS abbreviation.

Response: The full name of EDS has been added to the article, which you can see in line 139.

  1. Fig. 4 - please improve the quality of the drawing.

Response: The image quality has been optimized, you can view the updated Figure 6.

  1. Whether the stress-strain curves shown in Fig. 9 refer to individual samples or are they graphically averaged, please explain in the article.

Response: The stress-strain curve in the figure is the individual specimen stress-strain curve for each cycle number of powder deposited specimens. In this study, the deposited specimens from each cycle number powder were subjected to three tensile tests, and the data obtained from the experiments showed no significant difference in the stress-strain curves of the deposited specimens at the same cycle number, so the typical stress-strain curves of the new powder and the 4-cycle powder are given in the article.

  1. Do the 4 tensile strength values given in the application apply to single samples, or are they average values based on the measurement series performed?

Response: Tensile strength value is the average of three tensile tests for each deposited sample.

In addition, we have performed a fuller analysis of the powder particle size measurements, which you can see in the article on lines 121-129.

We tried our best to improve the manuscript and made some changes marked in red in revised paper which will not influence the content and framework of the paper. We appreciate for Editors/Reviewers’ work earnestly, and hope the correction will meet with approval. Once again, thank you very much for your comments and suggestions.

sincerely

Reviewer 2 Report

The paper presents the change in mechanical and some other metallurgical properties that occur when GH4169 alloy powder is recycled for additive manufacturing by laser melting deposition. The paper is well-presented and informative. However, the following significant changes are required to improve the quality of the paper.

1. The weight percentage of Nb is 5.4 in the elemental composition of GH4169 as presented in the Experiment section. However, in the abstract and in the analysis and discussion section, it is given as 4.31 wt%. This should be clarified.

2. The shape and size of the tensile specimens should be described. The number of tensile specimens used for each test condition should be mentioned as well.

Author Response

Dear Editor

We feel great thanks for your professional review work on our article. As you are concerned, there are several problems that need to be addressed. According to your nice suggestions, we have made extensive corrections to our previous draft, To make it easier for you to quickly find and view the changes, we have underlined the changes in red,the detailed corrections are listed below.

  1. The weight percentage of Nb is 5.4 in the elemental composition of GH4169 as presented in the Experiment section. However, in the abstract and in the analysis and discussion section, it is given as 4.31 wt%. This should be clarified.

Response: The elemental content of the powder in Table 1 is the theoretical value provided by the powder supplier Höganäs, the Nb elemental content is 5.4 wt%, the actual Nb elemental content of the powder measured in the experiment is 4.3 wt%, influenced by the powder manufacturing process, the actual value of the elemental content in the powder will have some differences with the ideal value.

  1. The shape and size of the tensile specimens should be described. The number of tensile specimens used for each test condition should be mentioned as well.

Response: As you suggest, the dimensions of the tensile specimens have been given in the article, which you can see in Figure 2 after line 111, and the number of tensile test specimens has been added to the article, which you can see in line 243.

We tried our best to improve the manuscript and made some changes marked in red in revised paper which will not influence the content and framework of the paper. We appreciate for Editors/Reviewers’ work earnestly, and hope the correction will meet with approval. Once again, thank you very much for your comments and suggestions.

sincerely

Reviewer 3 Report

The manuscript entitled “coatings-2280946” dealing with laser processing has been reviewed. The paper has been nicely written but needs significant improvement. Please follow my comments.

1.     Add a statement about the usage of your research in the coating.

2.     What is the main point of “Figure 5 “Metallographic photograph of powder deposition state”?

3.     What is the main issue that will be solved by this investigation? Please clarify it in the text.

4.     Please add a brief statement on your methodology in the abstract.

5.     What is the future direction of this work?

6.     Laser absorptivity is important which shows the quality of the parts and transition from keyhole to conduction mode. Please read and add the following ref in this area. “The effect of absorption ratio on meltpool features in laser-based powder bed fusion of IN718”.

7.     Please proofread the paper.

8.     Laser has many advantages over the conventional manufacturing method which can be highlighted in your paper. Please read the following manuscript and add it to the literature to show how the laser is comparable with conventional manufacturing.

·       Laser subtractive and laser powder bed fusion of metals: review of process and production features

Author Response

Dear Editor

We feel great thanks for your professional review work on our article. As you are concerned, there are several problems that need to be addressed. According to your nice suggestions, we have made extensive corrections to our previous draft, To make it easier for you to quickly find and view the changes, we have underlined the changes in red,the detailed corrections are listed below.

  1. Add a statement about the usage of your research in the coating.

Response: The usefulness of this study for depositing coatings on GH4169 alloy has been added to the article, which you can view in lines 74-78.

  1. What is the main point of “Figure 5 “Metallographic photograph of powder deposition state”?

Response: Figure 7 (original Figure 5) shows the metallographic microstructure of the new powder and the powder deposited state after four cycles. The main purpose of Figure 7 is to compare the differences in microstructure between the new powder and the powder deposited state after four cycles, and the results show that the powder deposited state specimens after four cycles of recycling have coarser dendrites compared with the new powder deposited state specimens, and the dendrite growth shows an overgrowth trend. The dendrites were extracted by ImageJ software and the percentage of dendrites in the same size area was measured 10 times for both powder deposited samples, and the results showed that the percentage of dendrites was more after 4 cycles of recovery.

  1. What is the main issue that will be solved by this investigation? Please clarify it in the text.

Response: The main questions to be addressed in this study have been added to the article, which you can view in lines 70-74.

  1. Please add a brief statement on your methodology in the abstract.

Response: A brief description of the study's research methods has been added to the abstract, which you can view in lines 7-12.

  1. What is the future direction of this work?

 Response: The future research directions of this study have three main points:

1)Optimize the powder screening method to obtain a more pure recovery powder in order to reduce the influence of impurities in the powder on the performance of the deposited samples.

2)GH4169 is an age-strengthened high-temperature alloy, and the effect of recycled powder on the deposited specimens after heat treatment will be investigated after this study.

3)This study is still a shallow study of the morphology of recycled powders, and the effect of the recycling process on powder characterization parameters such as powder bulk density, rest angle, collapse angle, and flat angle will be studied after this study.

  1. Laser absorptivity is important which shows the quality of the parts and transition from keyhole to conduction mode. Please read and add the following ref in this area. “The effect of absorption ratio on melt pool features in laser-based powder bed fusion of IN718”.

Response: We have read this literature that you have given. The effect of the change in laser absorbance on keyhole formation after recycling of the powder has been added based on the literature and in conjunction with the formation of pores in this study, which you can see in lines 228-233

  1.  Please proofread the paper.

Response: The article has been re-calibrated, you can check it again

  1. Laser has many advantages over the conventional manufacturing method which can be highlighted in your paper. Please read the following manuscript and add it to the literature to show how the laser is comparable with conventional manufacturing. “Laser subtractive and laser powder bed fusion of metals: review of process and production features”.

Response: We have read the literature that you have given us. We have added the advantages of laser additive manufacturing compared to traditional manufacturing methods based on the literature, which you can see in lines 36-41.

We tried our best to improve the manuscript and made some changes marked in red in revised paper which will not influence the content and framework of the paper. We appreciate for Editors/Reviewers’ work earnestly, and hope the correction will meet with approval. Once again, thank you very much for your comments and suggestions.

sincerely

Reviewer 4 Report

The topic chosen for the study is interesting and has manufacturing applications.

The authors poorly prepared the manuscript. The phrases “In this paper” should not be used, the title of the Introduction section is missed. Too many sentences that are too big and difficult for the reader to comprehend. English language and terminology needs to be corrected.

1 Introduction is too short.

2 The diagram in Figure 1 is hard to read, what does “Noble gas” mean? Usually shielding gas, or protective gas, or transporting gas is used.

3 Powder supply 25g/min is not too low?

4 How many beads were deposited?

5 Instead of "blocks" you need to use "samples".

6 Instead of corrosion solution, you need to use etchant.

7 I do not observe a significant difference between the photographs in figure 2.

8 What is meant by phrases “trace element”, “fresh powder”, “state tissure”.

Author Response

Dear Editor

We feel great thanks for your professional review work on our article. As you are concerned, there are several problems that need to be addressed. According to your nice suggestions, we have made extensive corrections to our previous draft, To make it easier for you to quickly find and view the changes, we have underlined the changes in red. Thanks for your suggestion. We have tried our best to polish the language in the revised manuscript. And we hope the revised manuscript could be acceptable for you. the detailed corrections are listed below.

  1. Introduction is too short.

Response: The introduction has been supplemented in part with an introduction to high-temperature alloys, the advantages of laser fusion deposition compared to conventional processing methods, two pieces of literature, the main purpose and role of this study, which you can view in the introduction section

  1. The diagram in Figure 1 is hard to read, what does “Noble gas” mean? Usually shielding gas, or protective gas, or transporting gas is used.

Response: We have changed the content of the diagram in Figure 1, the original "inert gas" refers to argon

  1. Powder supply 25g/min is not too low?

Response: The line speed used in this experiment is low, with a line speed of 1000 mm/min, and the deposition thickness is not high, with a deposition layer thickness of 1 mm, thus a powder feeding amount of 25 g/min is used in the experimental process. In the process of laser metal deposition, the powder feeding parameter mainly depends on the amount of the required deposition thickness, based on experimental experience, at the deposition speed of 1000mm/min, the powder feeding amount is generally between 15g/min and 30g/min according to the different deposition thickness.

  1. How many beads were deposited?

Response: We apologize for not specifying what you want to ask, if you are asking how many powder particles were deposited, we cannot give you an exact answer, we did not test the amount of powder particles used, but we can tell you that we consumed a total of 3kg of powder during the experiment.

  1. Instead of "blocks" you need to use "samples".

Response: As you suggested, all "blocks" have been replaced with "samples", which you can see in the article.

  1. Instead of corrosion solution, you need to use etchant.

Response: Corrosion solution has been replaced in the article, you can see it in line 108 of the article.

  1. I do not observe a significant difference between the photographs in figure 2.

Response: In order to more clearly express what we wanted to show in Figure 3, we have updated the content of Figure 3 (original Figure 2). You can see that the new powder of GH4169 (Figure 7a) has uniform size powder particles and smooth surface without adhesion, while as the number of cycles increases, by the 4th recycling, the powder particles have a clear satellite sphere morphology, with smaller size powder particles adhering around larger size powder particles.

  1. What is meant by phrases “trace element”, “fresh powder”, “state tissue”.

Response: "trace element" originally refers to the powder element content, has been replaced with "element", you can view in line 143; "fresh powder" originally refers to the new powder, has been replaced with "new powder", you can view in Figure 8a; "state tissue" originally refers to the deposited sample microstructure, has been replaced by "the microstructure of deposited samples", you can view in the 157th line.

We tried our best to improve the manuscript and made some changes marked in red in revised paper which will not influence the content and framework of the paper. We appreciate for Editors/Reviewers’ work earnestly, and hope the correction will meet with approval. Once again, thank you very much for your comments and suggestions.

sincerely

Round 2

Reviewer 1 Report

Dear Authors

I think that after all the corrections the article can be published.

Kind Regards

Reviewer

Reviewer 3 Report

The paper is in publishable format. 

Reviewer 4 Report

-